# Networks analysis of Brazilian climate data based on the DCCA cross-correlation coefficient

**Florêncio Mendes Oliveira Filho**[1,2]* , **Everaldo Freitas Guedes**[3] , **Paulo Canas Rodrigues**[4]

**1** Senai Cimatec University Center, Computer Engineering, Salvador, Brazil, **2** Earth Sciences and Environment Modeling Program, State University of Feira de Santana, Feira de Santana, BA, Brazil, **3** Climério de Oliveira Maternity School, Federal University of Bahia, Salvador, Brazil, **4** Department of Statistics, Federal University of Bahia, Salvador, Brazil

These authors contributed equally to this work.
* florenciofh@yahoo.com.br

**Data Availability Statement:** All relevant data are available at: https://portal.inmet.gov.br/dadoshistoricos. Others would be able to access these data in the same manner as the authors. The

## Abstract

Climate change is one of the most relevant challenges that the world has to deal with. Studies that aim to understand the behavior of environmental and atmospheric variables and the way they relate to each other can provide helpful insights into how the climate is changing. However, such studies are complex and rarely found in the literature, especially in dealing with data from the Brazilian territory. In this paper, we analyze four environmental and atmospheric variables, namely, wind speed, radiation, temperature, and humidity, measured in 27 Weather Stations (the capital of each of the 26 Brazilian states plus the federal district). We use the detrended fluctuation analysis to evaluate the statistical self-affinity of the time series, as well as the cross-correlation coefficient $\rho_{DCCA}$ to quantify the long-range cross-correlation between stations, and a network analysis that considers the top 10% $\rho_{DCCA}$ values to represent the cross-correlations between stations better. The methodology used in this paper represents a step forward in the field of hybrid methodologies, combining time series and network analysis that can be applied to other regions, other environmental variables, and also to other fields of research. The application results are of great importance to better understand the behavior of environmental and atmospheric variables in the Brazilian territory and to provide helpful insights about climate change and renewable energy production.

## Introduction

Due to its territorial immensity (8, 511, 965$Km^2$), with 93% of the territory located in the Southern Hemisphere and 7% in the Northern Hemisphere, with a location in the inter-tropical zone of the planet, Brazil has different types of climates, such as equatorial, tropical, highland tropical, humid tropical, semi-arid and subtropical. This distribution of climates, depending on the region, can have well-distributed rainfall (e.g., the South region of Brazil), high temperatures throughout the year, insufficient and poorly distributed rainfall (e.g., the

authors did not have any special access privileges that others would not have.

**Funding:** Florêncio Mendes Oliveira Filho is grateful to Centro Universitário SENAI CIMATEC and the National Council for Scientific and Technological Development (CNPq 25/2021) scholarship PDJ - 150655/2022-3. Paulo Canas Rodrigues would like to thank the National Council for Scientific and Technological Development of Brazil (CNPq) for the PQ - 2305852/2019 -1.

**Competing interests:** As described in the manuscript itself, the authors declare that there are no conflicting interests.

Northeast region of Brazil), high temperature, high humidity, and low thermal amplitude, with high levels of rainfall (e.g., the North region of Brazil, where the Amazon forest is located), mountain and plateau regions (e.g., the Southeast region of Brazil, which is characterized by low thermal amplitude and average temperature oscillating between $17°C$ and $22°C$) [1–9].

In countries with a vast territory, studies with climatic variables such as radiation, temperature, humidity and wind speed have been carried out to localize sandstorms in Asia [10]. This research explored the relationship between dust transported from various sources and ocean biological activities with different nutrient conditions and explores signs of global warming. Already in 2021, climate variables were investigated to explain hotter summers, frequent heavy rains, prolonged droughts, more severe forest fires, violent storms (blizzards), cyclones and the melting of the polar ice caps [11]. We also explore the relationship between climate variables (temperature, radiation, humidity and wind speed) and different aspects of the performance of buildings and energy systems [12–15]. In Brazil, the scalable behavior of wind speed along the coast was analyzed to determine long-range correlations and acquire more information about cross-sectional behavior at various scales. It was also investigated in Brazil with a focus on sustainable energy, variables such as radiation, temperature, and humidity, taking into account cross-correlation and multiple cross-correlations, by considering three stations located in the Brazilian state of Bahia [16]. Research with climate variables has also investigated significant energy sources in terms of their capacity, reliability, cost, safety, and effects on the environment [17]. For both studies, the concepts of auto-correlation and cross-correlation were applied to understand behavior, trends, and markings.

Regarding the topic of networks associated with climate variables, it has gained prominence in recent years. A study combines species distribution models with ecological network analysis to test the potential impacts of climate change on more than 700 plant and animal species in Central Europe's pollination and seed dispersal networks [18]. The networks have also been applied in studies to assess the impact of climate change on groundwater resources for regions without a pumping well. The network demonstrated the impact of climate change on average spring flow. Discharge increased by about +8% in spring and summer and decreased by about -7% in autumn and winter [19]. Network theory has also been used for the decision-making process in agriculture to effectively predict Fujian rice production under typical climatic conditions of the mountainous region and evaluate the performance of the network model against variations in development parameters by comparing the effectiveness of [20] multiple linear regression models to study soil water loss by evaporation and plant water loss by transpiration [21]. Regardless of the type of network, application and/or improvement of models expanded worldwide and in scientific domains.

Despite its vast territory and the importance of a better understanding of the behavior that climate variables such as wind speed, radiation, temperature, and humidity have in terms of climate change and renewable energy production, to the best of our knowledge, no study provides a portrait of the Brazilian regions using long-range cross-correlation and network analysis.

In this paper, we consider four important environmental variables: wind speed, radiation, temperature, and humidity, measured in 27 stations (the capital of each of the 26 Brazilian states plus the federal district). We use the detrended fluctuation analysis [22] to evaluate the statistical self-affinity of the signal and use the cross-correlation coefficient $\rho_{DCCA}$ [23] to quantify the long-range cross correlation between Brazilian regions. Then, network analysis is made for each environmental variable, where the top 10% values of $\rho_{DCCA}$ are highlighted and discussed.

The rest of the paper is organized as follows. The next section includes the materials and methods, where the data collection and organization are described, and the methodological

details of the detrended fluctuation analysis, DCCA cross-correlation coefficient, and network analysis. Then, the results for the detrended fluctuation analysis, DCCA cross-correlation analysis, and network analysis are discussed. The paper ends with some concluding remarks.

## Materials and methods

### Data collection and data organization

The data about the four environmental and atmospheric variables (global radiation, temperature, humidity, and wind speed) for all 27 Brazilian environmental stations (26 state capitals plus the federal district; Table 1) was extracted from the database of the Instituto Nacional de Pesquisas Espaciais—INPE (https://www.gov.br/inpe/pt-br).

In this study, 13 hourly observations (daily local time between 6 am to 6 pm; UTC 9 am to 9 pm), between January 1, 2010, and December 31, 2020 (4018 days; 52234 observations) were considered. Knowing that the original data is available per hour and per year for each of the variables studied, we joined the databases to compile the final time series data. The percentages of missing values in all 112 time series varied between 0.19% and 47.77%, being the missing values imputed by the mean of that hour along the days. However, [24] demonstrated that the DFA and DCCA methods are robust for these time series with missing values. The complete list of percentages of missing values is available in Table 1.

### Detrended fluctuation analysis

Peng et al. (1994) [22] developed the detrended fluctuation analysis (DFA) to analyze the existence of serial dependence (the statistical self-affinity of a signal), with the advantage of being also possible to be used in non-stationary data. Its main advantage is to avoid spurious detection of long-range dependence due to non-stationary data [25–29]. For a given "$Y$" time series, the algorithm is described as follows:

1. Define the cumulative sum ($X_N$) of the original signal fluctuations around its mean $\bar{Y}$):

$$X_N = \sum_{i=1}^{N} (Y_i - \bar{Y}) \tag{1}$$

2. Divide $X_N$ into boxes of equal length $n$.

3. In each box, fit the local trend of $X_n$ by a polynomial fit of degree ($n$), which represents the local trend of the box.

4. For the given $n$ box size, compute the detrended fluctuation function (root-mean-squared) of the signal $X_N$ as:

$$F(n) = \sqrt{\frac{1}{N} \sum_{i=1}^{N} (X_i - P(n))^2} \tag{2}$$

5. Step two is repeated for each of the available $\tau$ scales (box size) to obtain the empirical relationship between the overall fluctuation $F(n)$ and the box size $n$:

$$F(n) \propto n^{\alpha_{DFA}} \tag{3}$$

**Table 1. Region, state, the state capital, latitude, longitude, altitude, and percentage of missing values for each atmospheric variable for all 27 stations.** More details about the geographical location of each station can be found on the map in S1 Fig in S1 File of the supplementary material.

| Region | State | City (state capital) | Latitude | Longitude | Altitude | Percentage of missing values | | | |
|---|---|---|---|---|---|---|---|---|---|
| | | | | | | Radiation | Temperature | Humidity | Wind speed |
| North | PA | Belém | -1.41 | -48.44 | 24.00 | 8.65 | 1.18 | 1.20 | 1.21 |
| North | RR | Boa Vista | 2.82 | -60.69 | 94.00 | 23.44 | 12.13 | 12.13 | 12.21 |
| North | AP | Macapá | 3.81 | -51.86 | 21.00 | 36.95 | 32.19 | 32.20 | 32.23 |
| North | AM | Manaus | -3.10 | -60.02 | 61.25 | 11.98 | 1.77 | 1.90 | 8.58 |
| North | TO | Palmas | -10.18 | -48.30 | 280.00 | 11.12 | 5.20 | 5.20 | 5.20 |
| North | RO | Porto Velho | -8.76 | -63.47 | 98.00 | 32.22 | 24.76 | 24.77 | 47.77 |
| North | AC | Rio Branco | -9.96 | -68.17 | 220.00 | 44.81 | 37.00 | 42.19 | 37.03 |
| Northeast | SE | Aracaju | -10.95 | 37.05 | 4.72 | 1.90 | 1.33 | 12.02 | 12.68 |
| Northeast | CE | Fortaleza | -3.83 | -38.54 | 26.45 | 24.01 | 17.83 | 17.83 | 18.05 |
| Northeast | PB | João Pessoa | -7.17 | -34.82 | 47.00 | 11.15 | 11.04 | 11.89 | 10.59 |
| Northeast | AL | Maceió | -9.55 | -35.77 | 80.00 | 8.88 | 6.04 | 6.08 | 6.05 |
| Northeast | RN | Natal | -5.90 | -35.20 | 48.60 | 5.38 | 1.46 | 1.46 | 7.59 |
| Northeast | PE | Recife | -8.06 | -34.96 | 10.00 | 2.57 | 1.56 | 4.14 | 13.58 |
| Northeast | BA | Salvador | -13.02 | -38.52 | 51.41 | 3.35 | 1.95 | 1.95 | 1.98 |
| Northeast | MA | São Luís | -2.53 | -44.21 | 56.00 | 11.79 | 4.76 | 6.96 | 4.81 |
| Northeast | PI | Teresina | -5.07 | -42.81 | 74.36 | 9.52 | 7.90 | 7.90 | 7.91 |
| Central-West | DF | Brasília | -15.79 | -47.93 | 1159.54 | 6.20 | 0.35 | 0.35 | 0.89 |
| Central-West | MS | Campo Grande | -20.45 | -54.60 | 530.00 | 18.52 | 3.95 | 4.02 | 3.96 |
| Central-West | MT | Cuiabá | -15.56 | -56.73 | 240.00 | 26.67 | 13.14 | 13.14 | 16.93 |
| Central-West | GO | Goiania | -16.64 | -49.22 | 770.00 | 5.74 | 0.25 | 0.29 | 0.25 |
| Southeast | MG | Belo Horizonte | -19.88 | -43.97 | 869.00 | 4.73 | 0.21 | 0.21 | 0.19 |
| Southeast | RJ | Rio de Janeiro | -22.99 | -43.19 | 42.00 | 5.89 | 1.04 | 4.70 | 1.47 |
| Southeast | SP | São Paulo | -23.48 | -46.62 | 792.06 | 6.28 | 0.65 | 0.48 | 0.48 |
| Southeast | ES | Vitória | -20.27 | -40.30 | 90.00 | 6.86 | 4.01 | 5.22 | 4.02 |
| South | PR | Curitiba | -25.43 | -49.27 | 923.50 | 8.98 | 4.59 | 11.83 | 4.59 |
| South | SC | Forianopólis | -27.60 | -48.62 | 1.80 | 8.39 | 0.30 | 0.30 | 0.31 |
| South | RS | Porto Alegre | -30.05 | -51.17 | 46.97 | 8.10 | 0.39 | 0.39 | 1.33 |

The scaling exponent $\alpha_{DFA}$ quantifies the empirical strength of the long-range power-law correlations of the signal. If the signal is not random, it is characterized by long-range correlation features. The $\alpha_{DFA}$ can be interpreted as shown in Table 2 [30, 31].

The DFA method permits the detection of long-range correlations embedded in seemingly non-stationary time series, and also avoids the spurious detection of apparent long-range correlations, which are an artifact of non-stationarity.

**Table 2. Interpretation of the $\alpha_{DFA}$, a self-affinity parameter representing the long-range power-law correlation of the signal of a time series.**

| Exponent | Type of signal |
|---|---|
| $\alpha_{DFA} < 0.5$ | Antipersistent |
| $\alpha_{DFA} \simeq 0.5$ | Uncorrelated, white noise |
| $\alpha_{DFA} > 0.5$ | Long-range correlated persistent |
| $\alpha_{DFA} \simeq 1.0$ | 1/f noise |
| $\alpha_{DFA} > 1.0$ | Non-stationary |
| $\alpha_{DFA} \simeq \frac{3}{2}$ | Brownian noise |

### The DCCA cross-correlation coefficient

The DCCA cross-correlation coefficient proposed by Zebende (2011) [23] is centred on the ratio between the detrended covariance function $F^2_{DCCA}$ [32] and the variance function $F_{DFA}$ [22], as showed in the following five steps:

1. Consider two time series, $\{x_i\}$ and $\{y_i\}$, $i = 1, 2, \ldots, N$. These time series are integrated to obtain two new time series $\{X_k\}$ and $\{Y_k\}$:

$$X_k = \sum_{i=1}^{k}(x_i - \langle x \rangle) \quad \text{and} \quad Y_k = \sum_{i=1}^{k}(y_i - \langle y \rangle), \tag{4}$$

   where $\langle x \rangle$ and $\langle y \rangle$ represent the average of each time series.

2. These two integrated time series, $\{X_k\}$ and $\{Y_k\}$, are divided into $(N - n)$ overlapping boxes of equal length $n$, in which $n$ varies between 4 and $\frac{N}{4}$.

3. The local trend for each box is calculated by using a least-squares fit of each series, $\{\tilde{X}_{k,i}\}$ and $\{\tilde{Y}_{k,i}\}$, and the covariance of the residuals is calculated for each box:

$$f^2_{DCCA}(n, i) = \frac{1}{(n+1)}\sum_{k=i}^{i+n}(X_k - \tilde{X}_{k,i})(Y_k - \tilde{Y}_{k,i}). \tag{5}$$

   The variance of the residuals is also computed by using:

$$f^2_{DFA_x}(n, i) = \frac{1}{(n+1)}\sum_{k=i}^{i+n}(X_k - \tilde{X}_{k,i})^2 \tag{6}$$

$$f^2_{DFA_y}(n, i) = \frac{1}{(n+1)}\sum_{k=i}^{i+n}(Y_k - \tilde{Y}_{k,i})^2 \tag{7}$$

4. The average over all $(N - n)$ overlapping boxes is calculated to obtain the detrended covariance function:

$$F^2_{DCCA}(n) = \frac{1}{N-n}\sum_{i=1}^{N-n}f^2_{DCCA}(n, i), \tag{8}$$

   and the detrended variance functions:

$$F_{DFA_x}(n) = \sqrt{\frac{1}{N-n}\sum_{i=1}^{N-n}f^2_{DFA_x}(n, i)} \tag{9}$$

$$F_{DFA_y}(n) = \sqrt{\frac{1}{N-n}\sum_{i=1}^{N-n}f^2_{DFA_y}(n, i)} \tag{10}$$

5. Finally, the cross-correlation coefficient is computed as:

$$\rho_{(X_i, X_j)}(n) = \frac{F^2_{DCCA}(n)}{F_{DFA_x}(n)\ F_{DFA_y}(n)}, \tag{11}$$

where $\rho_{(X_i, X_j)}(n)$ ranges from $-1 \leq \rho_{(X_i, X_j)}(n) \leq 1$, in which:

i. $\rho_{(X_i, X_j)}(n) = 1$ means a perfect cross-correlation;

ii. $\rho_{(X_i, X_j)}(n) = 0$ there is no cross-correlation; and

iii. $\rho_{(X_i, X_j)}(n) = -1$ means a perfectly anti cross-correlation.

It is important to notice that $\rho_{(X_i, X_j)}$ depends on $n$ (time scale). So, one of the advantages of this detrended cross-correlation coefficient is to measure cross-correlations between two non-stationary time series at different time scales [24, 33–37].

## Network analysis

A graph is a mathematical structure that models the pairwise relations between objects. It includes vertices/nodes/points connected by edges/links/lines. There are two main types of graphs: (i) undirected graphs, where edges link two vertices without a specific direction, and (ii) directed graphs, where edges link two vertices in a specific direction. Formally, a graph can be defined as an ordered pair $G = (V, E)$, being $V$ a set of vertices and $E \subseteq \{\{x, y\} | x, y \in V, x \neq y\}$ a set of edges that connects two distinct edges $x$ and $y$. Network theory is related to the study of graphs in terms of symmetric or asymmetric relations between objects.

In this study, due to the nature of the variables, we consider undirected graphs and network analysis with symmetric relations between objects (in this case, the locations of the environmental variables). To conduct the network representation and analysis, we consider the DCCA cross-correlation coefficient between pairs of the Brazilian state capitals for each of the four environmental variables, temperature, humidity, radiation, and wind speed, for short median and long-distance correlations (box values of 13–one day, 1261–97 days, and 3328–256 days). The graph, based on the DCCA cross-correlation coefficients, was created using the function graph_from_adjacency_matrix of the R package igraph, and the network plot was obtained using the function ggnet2 of the R package GGally. To avoid overcrowded networks, we only depicted edges associated with the top 10% DCCA cross-correlation coefficients between objects.

## Results

In this section, we will include some of the main results and their discussion of the analysis of Brazilian climate data per state capital, for the DFA—detrended fluctuation analysis, DCCA cross-correlation analysis, and network analysis. Further tables and figures can be found in the supplementary material. General descriptive statistics for global radiation, temperature, air humidity, and wind speed, can be found in S1-S4 Tables in S1 File.

### Detrended fluctuation analysis (DFA)

We considered time series with 52.234 data points (hours), generated by the observation of 4.018 days (1/01/2010—31/12/2020), with 13 observations per day (daily local time between 6 am to 6 pm; UTC 9 am to 9 pm), for the variables radiation in $KJ/m^2$, maximum air temperature in $°C$ (dry bulb), the air humidity in %, and wind speed in $m/s$, and for the 26 Brazilian

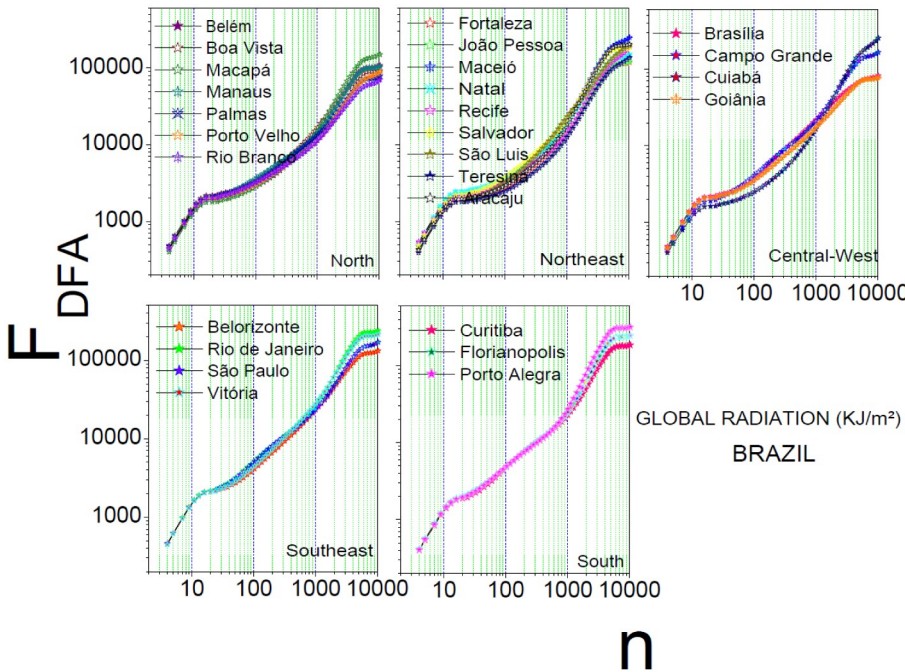

**Fig 1. Detrended fluctuation analysis for global radiation.** The plots show, respectively the curves for all capitals in the regions North, Northeast, Central-West, Southeast, and South, respectively. The vertical axis gives the $F_{DFA}$, and the horizontal axis shows the box size.

state capitals and the Brazilian capital, for a total of 108 time series. In this preliminary phase of the analysis, the objective was to evaluate the self-correlation of the variables in the Brazilian state capitals. The locations of the Brazilian states and regions can be seen in the S1 File, S1 Fig in S1 File, and the names of the state capitals with their location in Table 1. This results corroborates with studies of [24, 38–41].

Fig 1 shows the behavior of the fluctuation function $F_{DFA(n)}$ for global radiation in all state capitals per region. A characteristic pattern is observed for all time scales regardless of the region, with three clearly defined behaviors: (i) $n \leqslant 13$ (1 day and 6 scale observations: boxes with length 4, 5, 7, 9, 11, 13); (ii) $13 < n \leqslant 5625$ (1—430 days, with 48 scale observations: boxes with the length between 16 and 5625); and (iii) $n > 5625$ (430 days and 8 scale observations: boxes with length 6242, 6925, 7680, 8514, 943, 10457, 11585, 12831, 4327). We can also compute the $\alpha_{DFA}(medium) \cong 1.21(\pm 0.04)$ for up to 1 day of observations, the $\alpha_{DFA}(medium) \cong 0.87(\pm 0.02)$ for intervals between 1 and 430 days, and the $\alpha_{DFA}(average) \cong 0.06(\pm 0.01)$ for intervals greater than 430 days. $\alpha_{DFA}(medium) \cong 1.21$ represents a non-stationary persistent behavior, $\alpha_{DFA}(medium) \cong 0.87$ represents a long-range persistence, and $\alpha_{DFA}(medium) \cong 0.06$ represents an anti-persistent behavior.

With a behavior very similar to the global radiation, temperature (see S2 Fig in S1 File) also presents a specific pattern for $n \leq 13$, $13 < n \leq 5.625$, and $n > 5.625$. Only the state capitals in the Southeast and South regions showed a less intense transition in the trend associated with observations with $n \leq 13$, (1 day). Humidity (see S3 Fig in S1 File) in the state capitals for the Southeast and South regions and wind speed (S4 Fig in S1 File) in the state capitals for the Central-West, Southeast, and South regions, did not show the transition to $n \leq$ *and* 13, showing different characteristics for the state capitals in the North and Northeast regions.

Based on the analysis of the detrended fluctuation analysis of all environmental variables in all Brazilian state capitals, it can be concluded that the non-stationary and persistent behavior ($\alpha_{DFA}(medium) \cong 1.21$; $n \leqslant 13$), upward and/or downward trends are time dependent. In the long-range persistent behavior ($\alpha_{DFA}(medium) \cong 0.87$; $13 < n \leqslant 5.625$), it is observed that variables with closer values remain for a longer time, influencing the memory. As for the anti-persistent behavior ($\alpha_{DFA}(medium) \cong 0.06$; $n > 5.625$), a more stable behavior is expected as the scale reaches the maximum value.

It can be seen that the result of autocorrelation in the state capitals of the Southeast and South regions for humidity (S3 Fig in S1 File) and wind speed (S4 Fig in S1 File) have a different pattern due of their geographic locations (Table 1). The Southeast, with a tropical climate, has the greatest climate variability in Brazil, and its state capitals are influenced by elements such as altitude and latitude (Table 1) and air masses coming from the ocean. It is located in a transition area between an extremely dry region (Northeast region), with the altitude acting as a climate modeler distributing rainfall of different temperatures, and the South region. The South, on the other hand, with temperatures that are usually lower than the rest of Brazil, has a shorter summer with high temperatures and a rigorous winter, being a region that receives winds from the Southeast region, raising the humidity and heat for the region, increasing the temperature and causing rain.

**DCCA cross-correlation coefficient analysis.** In this section, to present the analysis of the $\rho_{DCCA}$ coefficient, the organization is done by Brazilian region (North, Northeast, Central-West, Southeast, and South) and by environmental variable (radiation, temperature, humidity, and wind speed). A detailed analysis of radiation is presented in this section, while the plots for the remaining variables are presented in the supplementary material: North (S5-S8 Figs in S1 File), Northeast (S8-S10 Figs in S1 File), Central-West (S11-S13 Figs in S1 File), Southeast (S14-S16 Figs in S1 File) and South (S17-S19 Figs in S1 File), for the environmental variables radiation, temperature, humidity and wind speed, respectively.

Fig 2 shows the results for global radiation in the North region. We can observe three characteristic behaviors, two maximum points at $n = 13$ (one day of observations—$\rho DCCA_{Medium} \sim 0.83$) and $n = 5625$ (430 days of observations—$\rho DCCA_{Medium} \sim 0.14$) and a minimum point at $n = 457$ (35 days of observations—$\rho DCCA_{Medium} \sim 0.70$), showing a variation in the fluctuation with the increase of the scale. With a minimum approaching zero for all correlations in six capitals in the North region (Belém, Macapá, Manaus, Palmas, Porto Velho, and Rio Branco), Boa Vista shows a different behavior with a minimum below zero ($\rho DCCA_{Palmas}$ x $\rho DCCA_{BoaVista}$), that is, a negative cross-correlation ($\rho_{DCCA} < 0$). The temperature (S5 Fig in S1 File) and humidity (S6 Fig in S1 File) show a behavior very similar to radiation. Once again, Boa Vista stands out in relation to the other capitals when analyzing temperature (Porto Velho, Belém, Macapá, and Palmas) and humidity for all capitals. For wind speed (S7 Fig in S1 File), up to $n = 13$, the cross-correlation coefficient is very similar, followed by individual variations per capital. The behavior was very similar for Manaus, Porto Velho, and Rio Branco for all scales, whereas Belém, Macapá, and Palmas show variations between $-0.73 < \rho_{DCCA} < 0.61$ for n > 457. These results corroborate with findings of [16, 42, 43].

Fig 3 shows the results for global radiation in the Northeast region. The $\rho_{DCCA}$ is positive for all capitals with a variation between $0.21 < \rho_{DCCA} < 0.88$, and the minimum value between ($\rho DCCA_{Teresina}$ x $\rho DCCA_{Maceio}$). S8 Fig in S1 File shows a positive cross-correlation for temperature between all capitals for $n \leq 135$ and with fluctuation between $-0.30 < \rho_{DCCA} < 0.99$ for all capitals after for values of $n$ above 135. Humidity (S9 Fig in S1 File) shows a positive cross-correlation ($\rho_{DCCA} > 0$) for all scales, with the exception of ($\rho DCCA_{Fortaleza}$ x $\rho DCCA_{Maceio}$) from $n > 3328$ (256 days) and ($\rho DCCA_{Teresina}$ x $\rho DCCA_{Maceio}$) to $n > 1570$ (120 days). The $\rho_{DCCA}$ coefficient shows a fluctuating between $-0.06 < \rho_{DCCA} < 0.91$ for all capitals. The wind

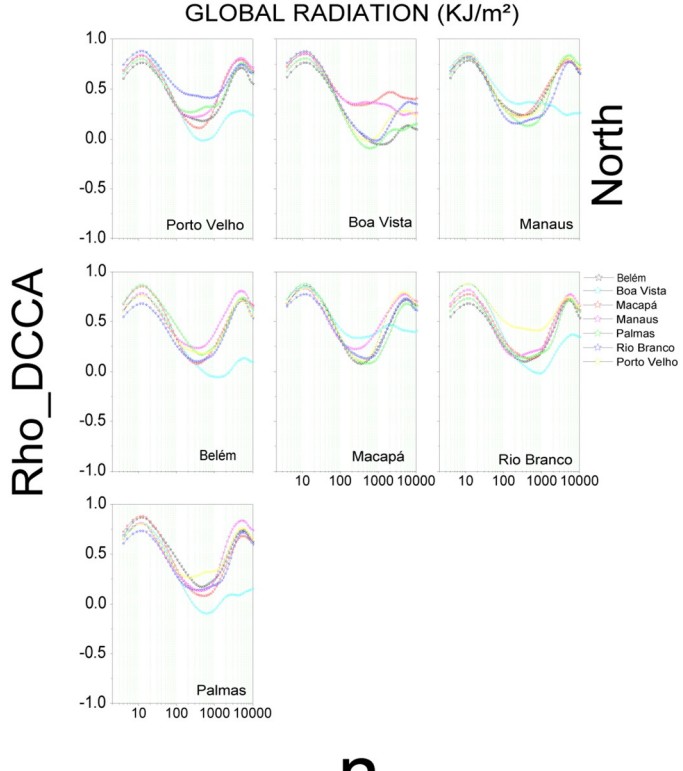

**Fig 2. Cross-correlation, $\rho_{DCCA}(n)$, for global radiation in the North region of Brazil.** The plots show the cross-correlations between the state capital written in the plot and all others in the region.

speed (S10 Fig in S1 File) showed a positive cross-correlation for all capitals, except for ($\rho DCCA_{Aracaju}$ x $\rho DCCA_{Recife}$) and ($\rho DCCA_{Aracaju}$ x $\rho DCCA_{JoaoPessoa}$) for $91 < n < 1261$, and ($\rho DCCA_{Teresina}$ x $\rho DCCA_{Aracaju}$) for $135 < n < 1261$. The $\rho_{DCCA}$ coefficient fluctuates between $-0.06 < \rho_{DCCA} < 0.92$ for all capitals.

In the Brazilian Central-West region (Fig 4 and S11-S13 Figs in S1 File, for radiation, temperature, humidity, and wind speed, respectively) showed positive cross-correlation ($\rho_{DCCA} > 0$) for all scales, being each variable with a specific fluctuation.

Fig 5 shows the results for global radiation in the Southeast region, with a positive cross-correlation ($\rho_{DCCA} > 0$) in all capitals. Maximum values are observed for $n = 13$ (one day) and $n = 5625$ (430 days), and minimum values are observed for $n = 252$ (19 days). Here the cross-correlation coefficient varies between $0.19 < \rho_{DCCA} < 0.81$. The temperature (S14 Fig in S1 File) also shows a positive cross-correlation coefficient for all capitals. With the increase of the scale for $n \leq 13$, we can see that the correlations tend to grow; however, for $13 < n \leq 38$, a decreasing trend in the correlations is visible, followed by growth for $n > 38$. For humidity (S15 Fig in S1 File), variations are visible, with a negative cross-correlation for large scales ($n > 4564$) for ($\rho DCCA_{Victory}$ x $\rho DCCA_{BeloHorizonte}$). For these four capitals, we observed a variation between $-0.08 < \rho_{DCCA} < 0.62$. Wind speed (S16 Fig in S1 File), shows a positive cross-correlation in São Paulo for all scales. Vitoria, Rio de Janeiro, and Belo Horizonte in the range between $45 < n < 1130$ show negative $\rho_{DCCA}$ values. In these four capitals, a variation between $-0.19 < \rho_{DCCA} < 0.69$ is observed.

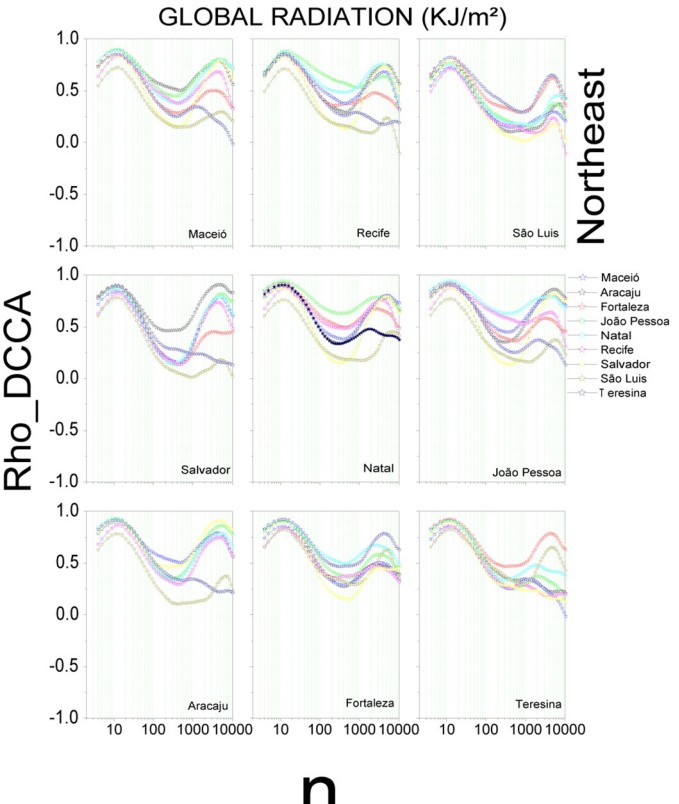

**Fig 3. Cross-correlation, $\rho_{DCCA}(n)$, for global radiation in the Northeast region of Brazil.** The plots show the cross-correlations between the state capital written in the plot and all others in the region.

Fig 6 shows the results for global radiation in the South region. For small scales ($n \leq 13$) an increasing trend is observed, followed by a decrease for $13 < n \leq 576$, and by another increase for $n > 576$. Here the correlation varies between $0.18 < \rho_{DCCA} < 0.99$. For temperature (S17 Fig in S1 File), after $n > 38$ (3 days), increasing the scale also increases the cross-correlation. For these three capitals, a variation between $0.38 < \rho_{DCCA} < 0.99$ is observed. Humidity (S18 Fig in S1 File) and wind speed (S19 Fig in S1 File), some stability between $24 < n < 809$ was observed, being more evident for humidity in Florianópolis ($0.13 < \rho_{DCCA} < 0.22$) and for the wind speed in Curitiba, Florianópolis and Porto Alegre with the cross-correlation coefficient varying between $0.17 < \rho_{DCCA} < 0.20$. Differently from other regions, the cross-correlation coefficient tends to fall for wind speed when considering $n > 5068$, with a variation between $0.75 < \rho_{DCCA} < 0.39$.

## Network analysis

In this third phase of our analysis, we use the concept of network analysis to recognize hidden patterns in the data and to group and classify them. Our analysis takes into account the stations located in each Brazilian state capital and Federal District (Table 1). The nodes of the interconnected networks are represented by the capitals, and the edges are represented by the strength of the cross-correlation coefficient, which satisfies $\rho_{DCCA} \geq 90\%$. The choice of a threshold of 90% takes into account the representation of strong correlations and also avoids overcrowding in the representation of networks. The networks shown in Figs 7–10, are associated to the variables radiation, temperature, humidity and wind speed, respectively. The criteria adopted

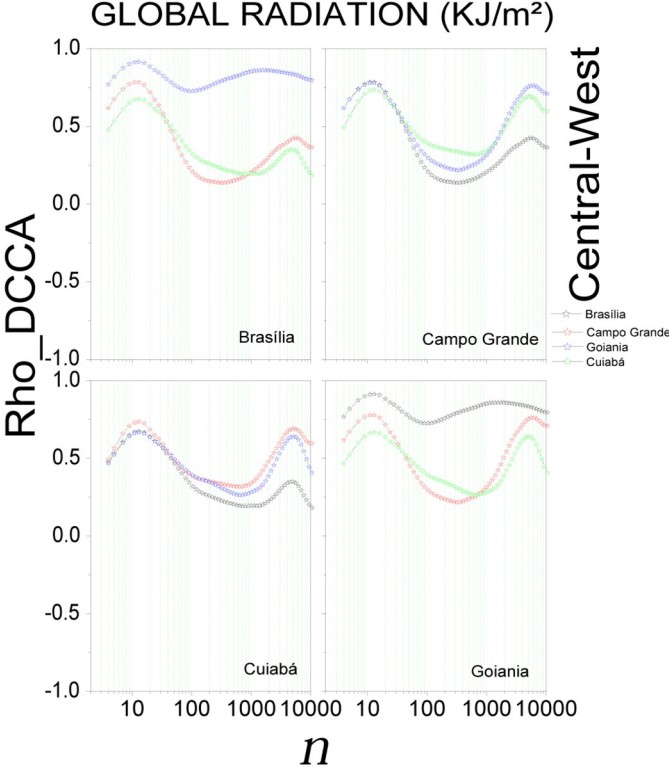

**Fig 4. Cross-correlation, $\rho_{DCCA}(n)$, for global radiation in the Central-West region of Brazil.** The plots show the cross-correlations between the state capital written in the plot and all others in the region.

form boxes of sizes $n = 13$ (one day of observations), $n = 1261$ (97 days of observations), and $n = 3328$ (256 days of observations).

Fig 7 shows the network analysis for solar radiation. Regardless of the box size (13, 1261, and 3328), we see subgroups of networks interconnected mostly by nearby regions. For $n = 13$, the only interconnected network subgroup shows a link between seven Northeast capitals, two Central-West capitals, and one Southeast capital. When considering $n = 1261$, this sub-network also includes the three Southeast capitals that border the Atlantic Ocean and the three South capitals that also border the ocean. For $n = 3328$ some of the Northeast capitals closer to the equator are removed from the sub-network.

Fig 8 shows the network analysis for the air temperature. When analyzing the box size $n = 13$, we see a strong cross-correlation, however, fragmented into three subgroups of networks, the first represented by the capitals of Rio Grande do Norte, Ceará, Maranhão and Pará, from the North and Northeast regions; the second formed by the capitals of Alagoas, Pernambuco, and Paraíba, in the Northeast; and the third by the capitals of Minas Gerais, Tocantins, Goiás, Federal District, Piauí, Amapá, Roraima, Rondônia, Acre, Mato Grosso, and Mato Grosso do Sul. For $n = 1261$, three subgroups are also formed: (i) the capitals of Mato Grosso and Mato Grosso do Sul, with hot and humid climates; (ii) the capitals of the Northeast states of Bahia, Sergipe, Pernambuco, Rio Grande do Norte, Alagoas and Paraíba, all of them bordering the Atlantic Ocean; and the capitals of Rio Grande do Sul, Santa Catarina, Paraná, São Paulo, Rio de Janeiro, Espírito Santo, Minas Gerais, Distrito Federal and Goiás, states located in a close geographic location between the South, Southeast and Central-West regions. For $n = 3328$, two subgroups are formed, the first by the capital of Goiás and the Distrito Federal,

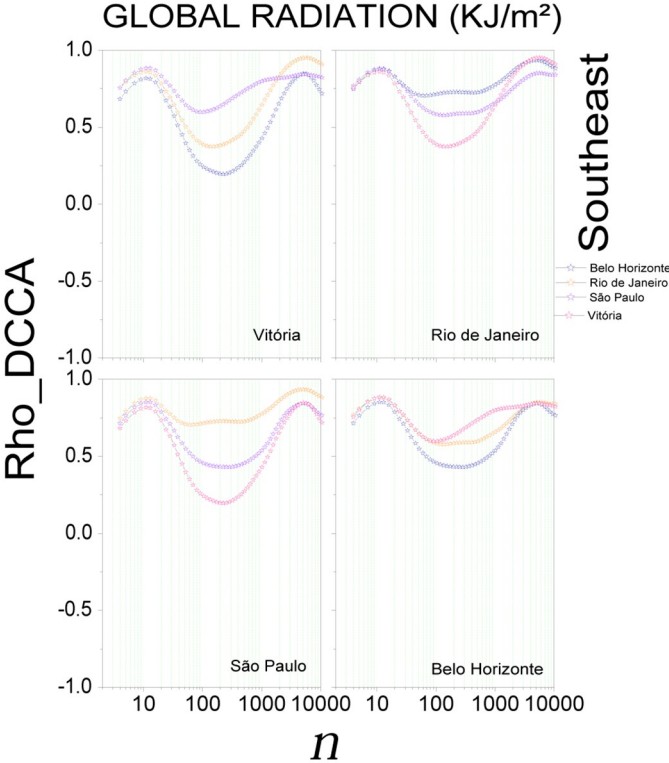

**Fig 5. Cross-correlation, $\rho_{DCCA}(n)$, for global radiation in the Southeast region of Brazil.** The plots show the cross-correlations between the state capital written in the plot and all others in the region.

which are very near geographically, and the second by the capitals of Pernambuco, Paraíba, Sergipe, Alagoas, Rio Grande do Norte, Bahia, Rio de Janeiro, São Paulo, Espirito Santo, Santa Catarina, Rio Grande do Sul and Paraná, representing almost all Brazilian coast from the Northeast to the South.

Figs 9 and 10 show the network analysis for the relative air humidity and wind speed, respectively, and their interpretation can be done in a similar manner as in Figs 7 and 8. Similarly to radiation and air temperature, the edges are mostly connections between neighboring regions and regions located on the Brazilian coast.

## Discussion

In this paper, we seek to analyze the time series of radiation, temperature, humidity, and wind speed from 27 Brazilian weather stations, in five regions, from the point of view of autocorrelation and cross-correlation. Throughout Detrended fluctuation analysis (DFA), we revealed a characteristic pattern for radiation for all scales, regardless of the region. For temperature, similarly to radiation, we observed a less intense transition for the Southeast and South regions. For humidity in the Southeast and South regions and wind speed in the Midwest, Southeast, and South regions, there was no transition for one day of observation (n = 13), and different characteristics for state capitals in the North and Northeast regions were observed. Persistent and anti-persistent behavior was interpreted as time variations, with close values influencing the memory of observations (hour-to-hour). Autocorrelation results for Southeast and South

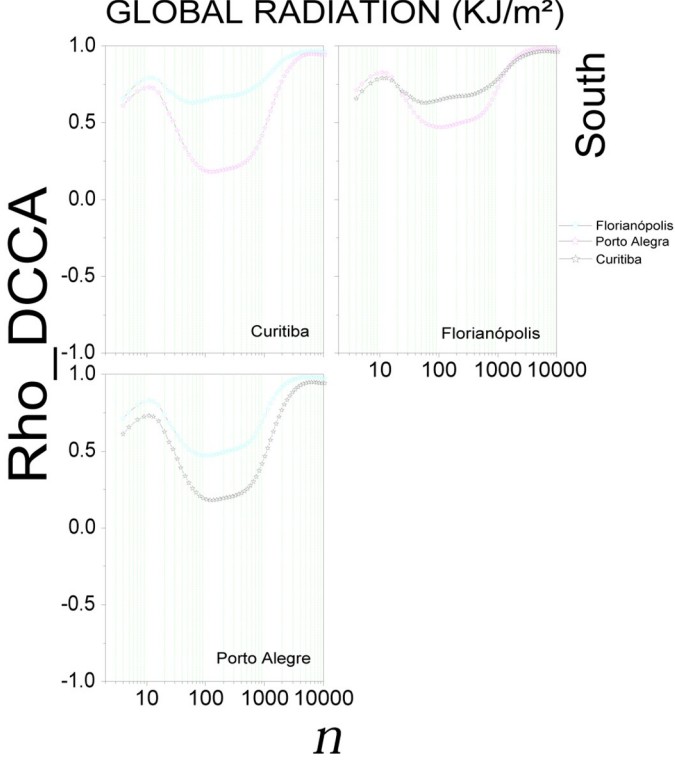

**Fig 6. Cross-correlation, $\rho_{DCCA}(n)$, for global radiation in the South region of Brazil.** The plots show the cross-correlations between the state capital written in the plot and all others in the region.

states for humidity and wind speed revealed differences due to geographic location and local climate.

The approach presented here via DFA follows the same line of thought as the Climatic Standards presented in the 24/03/2022 edition of (INMET) on meteorological variables of the last

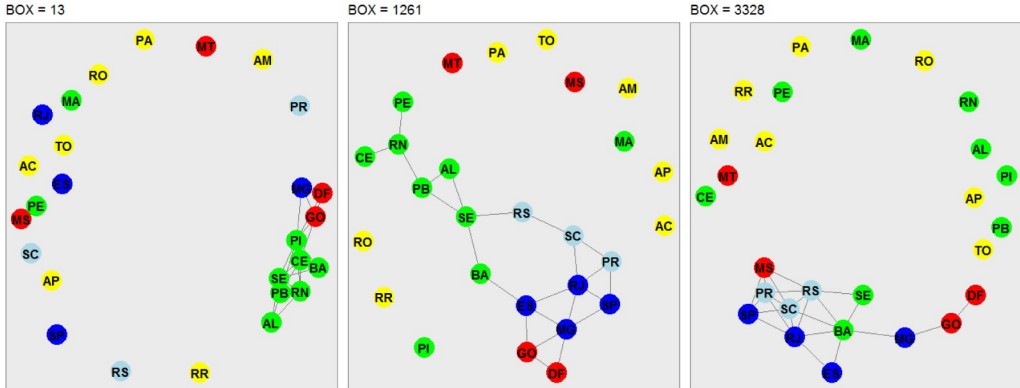

**Fig 7. Network representation of the environmental variable global solar radiation, considering the 27 Brazilian state capitals as vertices and the edges defined by the DCCA cross-correlation coefficient between pairs of Brazilian state capitals.** Only edges associated with the top 10%$DCCA$ cross-correlation coefficients between objects were depicted. The five colors in the vertices correspond to the state capitals in each of the five Brazilian regions. Box sizes of 13–one day, 1261–97 days, and 3328–256 days are depicted, from left to right.

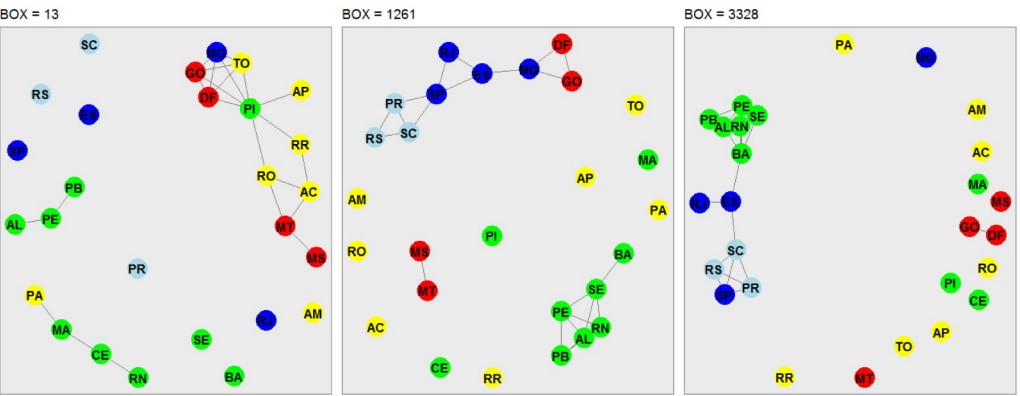

**Fig 8. Network representation of the environmental variable air temperature, considering the 27 Brazilian state capitals as vertices and the edges defined by the DCCA cross-correlation coefficient between pairs of Brazilian state capitals.** Only edges associated with the top 10% DCCA cross-correlation coefficients between objects were depicted. The five colors in the vertices correspond to the state capitals in each of the five Brazilian regions. Box sizes of 13–one day, 1261–97 days, and 3328–256 days are depicted, from left to right.

decades (https://portal.inmet.gov.br/). An advantage of the method used in this study, when compared to the average values of INMET in the period between 01/01/1991 and 31/12/2020, is the self-affinity of the scale. While INMET computes the mean variables from month to month, the DFA computes hour-to-hour variations for small, medium, and large scales. The contributions not only raise the possibility of discussions associated with scientific communication but can also be useful to understand and collaborate with various economic activities, agribusiness, the energy generation sector (hydraulic, wind, and/or solar), sports activities and leisure, and, above all, urban planning, among others.

Regarding the cross-correlation coefficient $\rho_{DCCA}$, we observed the influence of the correlation by region. Important aspects were verified by three characteristic behaviors in the scale and variation in the fluctuation by region. This type of correlation study was also verified by

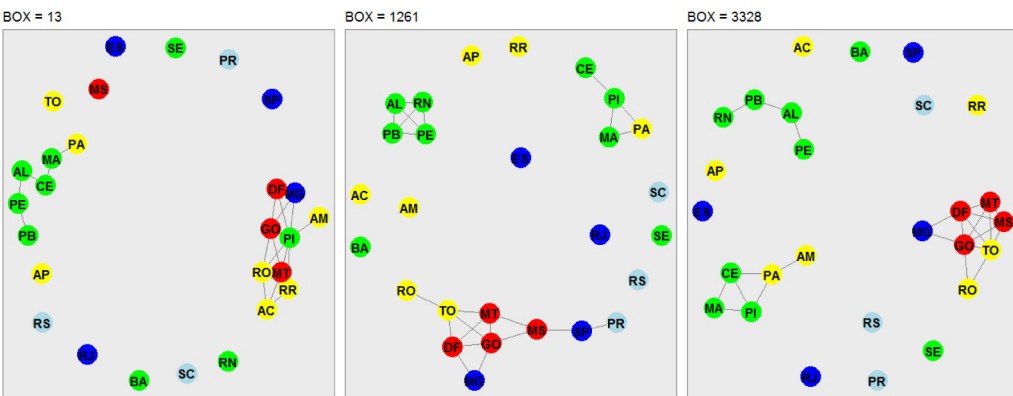

**Fig 9. Network representation of the environmental variable relative air humidity, considering the 27 Brazilian state capitals as vertices and the edges defined by the DCCA cross-correlation coefficient between pairs of Brazilian state capitals.** Only edges associated with the top 10% DCCA cross-correlation coefficients between objects were depicted. The five colors in the vertices correspond to the state capitals in each of the five Brazilian regions. Box sizes of 13–one day, 1261–97 days, and 3328–256 days are depicted, from left to right.

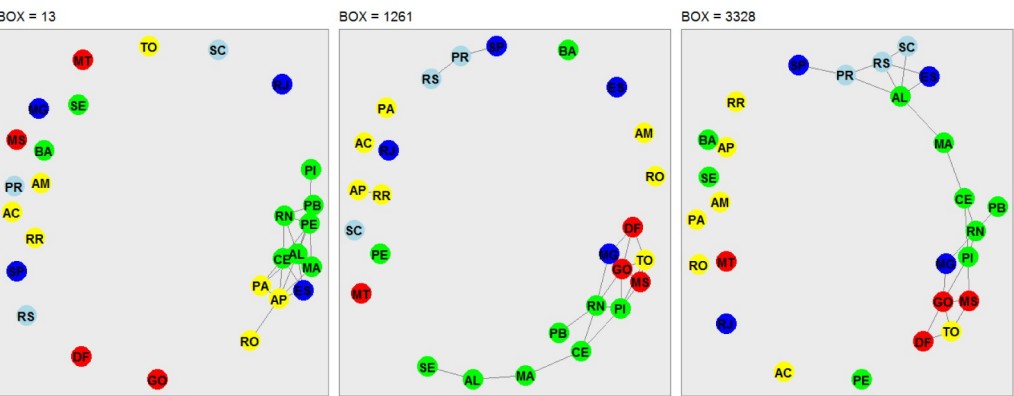

**Fig 10. Network representation of the environmental variable wind speed, considering the 27 Brazilian state capitals as vertices and the edges defined by the DCCA cross-correlation coefficient between pairs of Brazilian state capitals.** Only edges associated with the top 10% DCCA cross-correlation coefficients between objects were depicted. The five colors in the vertices correspond to the state capitals in each of the five Brazilian regions. Box sizes of 13–one day, 1261–97 days, and 3328–256 days are depicted, from left to right.

Brito [16], specifically with the Extended Multiple Correlation Coefficient $DMC_2^x$ applied to radiation, temperature, and relative humidity between three stations in the interior of the Bahia. As with INMET, Brito [16] also mentioned the importance of this type of study as a contribution to the energy sector.

Finally, we show as a highlight for this study the visualization between regions through network analysis, with the cross-correlation coefficient roDCCA representing the edges measuring the intensity of the correlation between the 26 capitals and the Federal District. Unlike traditional techniques, such as average degree, weighted average degree, network diameter, modularity, PageRank, and connected components, among others, the interconnected networks were made by boxes representing sizes $n = 13$ (one day of observations), $n = 1261$ (97 observation days) and $n = 3328$ (256 observation days) varied. Another relevant detail in this work was the representation of the networks in which we sought a non-polluting visualization graphically and that the model employed showed the most correlated regions for the variables. However, we opted for regions with a strong cross-correlation of 90%. This does not prevent replicating the methodology for other correlation percentages.

## Concluding remarks

To better understand the behavior of the Brazilian climate, in this paper, four environmental and atmospheric variables, wind speed, radiation, temperature, and humidity, were measured in 27 stations (the capital of each of the 26 Brazilian states plus the federal district) were considered. The detrended fluctuation analysis was used to evaluate the statistical self-affinity of the signal being the overall main conclusion that the non-stationary and persistent behavior ($\alpha_{DFA}(medium) \cong 1.21$; $n \leqslant 13$) upward and/or downward trends are time dependent. In the long-range persistent behavior ($\alpha_{DFA}(medium) \cong 0.87$; $13 < n \leqslant 5.625$), it is observed that variables with closer values remain for a longer time, influencing the memory. As for the anti-persistent behavior ($\alpha_{DFA}(medium) \cong 0.06$; $n > 5.625$), a more stable behavior is expected as the scale reaches the maximum value.

The cross-correlation coefficient $\rho_{DCCA}$ was used to quantify the long-range cross-correlation between Brazilian state capitals per geographical region (North, Northeast, Central-West,

Southeast, and South) and per atmospheric variable. Similar behavior for the $\rho_{DCCA}$ were found when considering state capitals near to each other geographically.

The network analysis was based on the DCCA cross-correlation coefficient ($\rho_{DCCA}$) between pairs of the Brazilian state capitals for each of the four environmental variables. Short, medium, and long-distance correlations (box values of 13–one day, 1261–97 days, and 3328–256 days) were considered, and only values of the top 10% DCCA cross-correlation coefficients were depicted in the networks. The edges of the networks for each of the atmospheric variables and each box size are mostly connections between neighboring regions and regions located on the Brazilian coast.

The methodology used in this paper represents a step forward in the field of hybrid methodologies, combining time series and network analysis that can be applied to other regions, other environmental variables, and also to other fields of research. The results presented are of great importance to better understand the behavior of environmental and atmospheric variables in Brazil, their long-range cross-correlation, and the way they relate to each other, and provide helpful insights into how the climate is changing and renewable energy production.

## Supporting information

**S1 File. Supplementary material with map, graphs and tables used in the analysis of climate variables (Temperature, Radiation, Humidity and Wind Speed).**
(PDF)

## Author Contributions

**Conceptualization:** Florêncio Mendes Oliveira Filho, Everaldo Freitas Guedes.

**Data curation:** Florêncio Mendes Oliveira Filho, Everaldo Freitas Guedes, Paulo Canas Rodrigues.

**Formal analysis:** Everaldo Freitas Guedes, Paulo Canas Rodrigues.

**Funding acquisition:** Florêncio Mendes Oliveira Filho, Everaldo Freitas Guedes.

**Investigation:** Florêncio Mendes Oliveira Filho, Everaldo Freitas Guedes.

**Methodology:** Florêncio Mendes Oliveira Filho, Everaldo Freitas Guedes, Paulo Canas Rodrigues.

**Project administration:** Florêncio Mendes Oliveira Filho, Paulo Canas Rodrigues.

**Resources:** Florêncio Mendes Oliveira Filho, Everaldo Freitas Guedes.

**Software:** Florêncio Mendes Oliveira Filho, Everaldo Freitas Guedes, Paulo Canas Rodrigues.

**Supervision:** Florêncio Mendes Oliveira Filho, Paulo Canas Rodrigues.

**Validation:** Florêncio Mendes Oliveira Filho, Everaldo Freitas Guedes, Paulo Canas Rodrigues.

**Visualization:** Florêncio Mendes Oliveira Filho, Everaldo Freitas Guedes, Paulo Canas Rodrigues.

**Writing – original draft:** Florêncio Mendes Oliveira Filho, Everaldo Freitas Guedes, Paulo Canas Rodrigues.

**Writing – review & editing:** Florêncio Mendes Oliveira Filho, Everaldo Freitas Guedes, Paulo Canas Rodrigues.

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
