## [Decision Letter · Decision Letter 0]

18 Jun 2023

PONE-D-23-12144Networks Analysis of Brazilian Climate Data based on the

DCCA cross-correlation coefficientPLOS ONE

Dear Dr. Oliveira Filho,

Thank you for submitting your manuscript to PLOS ONE. After careful consideration, we feel that it has merit but does not fully meet PLOS ONE’s publication criteria as it currently stands. Therefore, we invite you to submit a revised version of the manuscript that addresses the points raised during the review process.

We look forward to receiving your revised manuscript.

Kind regards,

Pak Wai Chan

Academic Editor

PLOS ONE

“Florˆencio Mendes Oliveira Filho thanks National Council for Scientific and

Technological Development (CNPq 25/2021) ”bolsa de PDJ’- Process: 150655/2022-3,

Brazilian agency. Paulo Canas Rodrigues acknowledges financial support from the

Brazilian National Council for Scientific and Technological (CNPq) grant ”bolsa de

produtividade PQ-2”305852/2019-1.”

“O primeiro autor do artigo é docente do Centro Universitário Senai Cimatec. Neste sentido, o Centro Unitário  estará financiando os custos da publicação.”

5. We noted in your submission details that a portion of your manuscript may have been presented or published elsewhere. [The research was submitted to Scientific Report (Peer Review Advisors) on 05 February 2022 UTC (Submission ID cd038b47-9e6e-42d6-9f25-5d2aac8cffd3). On 05/18/2022 we received the article for correction. On 05/11/2022 the article was rejected.

In the submission process to Scientific Report, the article adhered to the Research Square policy that was registered (DOI https://doi.org/10.21203/rs.3.rs-1330103/v1.).

After the rejection of "Scientific Reports – Nature" the article was submitted to Jornal do Clima and could not be continued (decision of the Editor-in-Chief Dr. Mingfang Ting) because our article is still being published in an environment linked to Scientific Reports, in this case the Research Square DOI.

In contact with Nusrat Tasneem (Editorial Support in Scientific Reports), the information given is that the DOI is no longer updated as soon as it is accepted by a journal.] Please clarify whether publication was peer-reviewed and formally published. If this work was previously peer-reviewed and published, in the cover letter please provide the reason that this work does not constitute dual publication and should be included in the current manuscript.

6. We note that you have indicated that data from this study are available upon request. PLOS only allows data to be available upon request if there are legal or ethical restrictions on sharing data publicly. For more information on unacceptable data access restrictions, please see http://journals.plos.org/plosone/s/data-availability#loc-unacceptable-data-access-restrictions.

7. We note that Figure 1S in your submission contain [map/satellite] images which may be copyrighted. All PLOS content is published under the Creative Commons Attribution License (CC BY 4.0), which means that the manuscript, images, and Supporting Information files will be freely available online, and any third party is permitted to access, download, copy, distribute, and use these materials in any way, even commercially, with proper attribution. For these reasons, we cannot publish previously copyrighted maps or satellite images created using proprietary data, such as Google software (Google Maps, Street View, and Earth). For more information, see our copyright guidelines: http://journals.plos.org/plosone/s/licenses-and-copyright.

a. You may seek permission from the original copyright holder of Figure 1S to publish the content specifically under the CC BY 4.0 license. 

Additional Editor Comments:

In view of the comments from the reviewers, major revision of the manuscript would be required.

Reviewers' comments:

Reviewer's Responses to Questions

**Comments to the Author**

1. Is the manuscript technically sound, and do the data support the conclusions?

Reviewer #1: Yes

Reviewer #2: Partly

2. Has the statistical analysis been performed appropriately and rigorously? 

Reviewer #1: Yes

Reviewer #2: Yes

3. Have the authors made all data underlying the findings in their manuscript fully available?

Reviewer #1: Yes

Reviewer #2: No

4. Is the manuscript presented in an intelligible fashion and written in standard English?

Reviewer #1: Yes

Reviewer #2: No

5. Review Comments to the Author

Reviewer #1: Thank you for submitting your paper “Análise de redes de dados climáticos brasileiros com base no coeficiente de correlação cruzada DCCA” to the Journal of PLOS ONE.

In this paper, authors analyze four environmental and atmospheric variables, namely, wind speed, radiation,

temperature, and humidity, measured in 27 Weather Stations (the capital of each of the 26 Brazilian states plus the federal district).

The article is well written and well structured as a scientific text; the literature review is comprehensive and up to date. However, several issues need to be addressed properly before the paper is considered for publication.

Add the highlights

Add a table of nomenclature

Improve introduction and references

https://doi.org/10.1016/j.jobe.2021.103057

https://doi.org/10.1016/j.energy.2021.121641

https://doi.org/10.1016/j.enconman.2022.116554

https://doi.org/10.1016/j.jclepro.2023.137345

Figure 2S to 19S, Replace green vertical lines with black or different shades of gray for better readability.

In conclusion please explain how your work advances the present state of knowledge. Please, be more effective in conclusion paragraph. Critical discussion about the applicability and weak points of the proposed approach will make this manuscript much better.

Reviewer #2: relevant challenges that the world has to deal with.- what do you mean ‘relevant’ challenges ?

Line 73-74 deal with being the missing values imputed by the mean of that hour along

the days. –elaborate how it is done

However, [20] demonstrated that the DFA and DCCA methods are… make it more details are required 47 % missing means .. still DCCA works ? what exactly authors mean is not clear

Tabela 2.- Figura 1. The paper should be completely in English

Throughout the paper, the use of symbols should be checked For example kJ ; Line 223; Figure 2 etc in many parts

A proper connection between the three analysis performed is to be conveyed by a flowchart ;better clarity is required for line 130 onwards

IN Figures of DCCA, some of the lines are not visible ..proper color selection is required

The discussion section, especially last paragraph should be redrafted ..also what is the usefulness of the work/this graphical representations is not clear .. moreover the statements of climate change etc are vague

6. PLOS authors have the option to publish the peer review history of their article (what does this mean?). If published, this will include your full peer review and any attached files.

Reviewer #1: No

Reviewer #2: No

---

## [Author Response · Author response to Decision Letter 0]

14 Aug 2023

REPLY TO REVIEWERS - PLOS ONE [PONE-D-23-12144]

Magazine requirements:

When submitting your review, we need you to meet these additional requirements.

1. Make sure your manuscript meets the PLOS ONE style requirements, including those for file naming. PLOS ONE style templates can be found at

Answer: We certify and verify that the manuscript is in the Plos One model.

2. Please note that PLOS ONE has specific guidelines on code sharing for submissions where author-generated code supports manuscript findings. In such cases, all code generated by the author must be made available without restrictions at the time of publication of the work. Please review our guidelines at https://journals.plos.org/plosone/s/materials-and-software-sharing#loc-sharing-code and ensure that your code is shared in a way that follows best practices and facilitates reproducibility and reuse.

Answer: Code provided.

Just information about our work base. The data are public and accessible to any researcher.

Figure 1S Code: Replaced in Supplementary Material.

#library(devtools)

library(geobr)

library(ggplot2)

#library(sf)

#library(dplyr)

# Available data sets

datasets <- list_geobr()

head(datasets)

# State of Sergige

state <- read_state(

 code_state="SE",

 year=2020,

 showProgress = FALSE

)

# Municipality of Sao Paulo

muni <- read_municipality(

 code_muni = 3550308, 

 year=2010, 

 showProgress = FALSE

)

plot(muni['name_muni'])

# read all intermediate regions

inter <- read_intermediate_region(

 year=2020,

 showProgress = FALSE

)

# read all states

states <- read_state(

 year=2017, 

 showProgress = FALSE

)

head(states)

# Remove plot axis

no_axis <- theme(axis.title=element_blank(),

 axis.text=element_blank(),

 axis.ticks=element_blank())

lat = c(-10.83, -8.77, -3.47, 1.99, -3.79, 1.41, -9.46, -5.42, -6.6, -5.2, -5.81, -7.28, -8.38, -9.62, -10.57, -13.29, -18.1, -19.19, -22.25, -22.19, -24.89, -27.45, -30.17, -20.51, -12.64, -15.98, -15.83)

long = c(-63.34, -70.55, -65.1, -61.33, -52.48, -51.77, -48.26, -45.44, -42.28,-39.53, -36.59, -36.72, -37.86, -36.82, -37.45,-41.71, -44.38,-40.34, -42.66, -48.79, -51.55, -50.95, -53.5, -54.54, -55.42, -49.86, -47.86)

lab =c("RO", "AC", "AM", "RR", "PA", "AP", "TO", "MA", "PI", "CE", "RN", "PB", "PE", "AL", "SE", "BA", "MG", "ES","RJ", "SP", "PR", "SC", "RS", "MS", "MT","GO", "DF")

# Plot all Brazilian states

ggplot() +

 geom_sf(data=states, aes(fill=name_region), color= 'gray', size=.15)+ 

 labs(subtitle="Brazilian regions and states", size=8) +

 theme(panel.background = element_rect(fill = "white", colour = "grey50"), 

 legend.title = element_blank(),

 axis.title.x = element_blank(),

 axis.title.y = element_blank())+

 scale_fill_manual(values = c("red", "darkgreen", "green", "blue", "magenta"),

 labels = c("Central-West", "Northeast", "North", "Southeast", "South"))+

annotate("text", x=long, y=lat, label=lab, size=4)

Network code:

rm(list=ls())

setwd("C:\\\\Users\\\\toparo\\\\Desktop\\\\Florencio e Everaldo\\\\Networks")

# library

library(igraph)

###########

# BOX = 13 

############

cor.radiation.1<- read.csv("C:\\\\Users\\\\toparo\\\\Desktop\\\\Florencio e Everaldo\\\\Networks\\\\radiation_13.csv", sep=";", h=T)[,-1]

# names(cor.radiation.1)<- c("SE", "CE", "PB", "AL", "RN", "PE", "BA", "MA", "PI", "PA",

# "AP", "TO", "AC", "RR", "AM", "RO", "DF", "MS", "GO", "MT",

# "MG", "RJ", "SP", "ES", "PR", "SC", "RS")

names(cor.radiation.1)<- c("SE", "PA", "MG", "RR", "DF", "MS", "MT", "PR", "SC", 

 "CE", "GO", "PB", "AP", "AL", "AM", "RN", "TO", "RS",

 "RO", "PE", "AC", "RJ", "BA", "MA", "SP", "PI", "ES")

col<- c("green", "yellow", "blue", "yellow", "red", "red", "red", "lightblue", "lightblue", 

 "green", "red", "green", "yellow", "green", "yellow", "green", "yellow", "lightblue",

 "yellow", "green", "yellow", "blue", "green", "green", "blue", "green", "blue")

# c(rep("green",9), -- Nordest ("SE", "CE", "PB", "AL", "RN", "PE", "BA", "MA", "PI")

# rep("yellow", 7), -- North ("PA", "AP", "TO", "AC", "RR", "AM", "RO")

# rep("red", 4), -- Central west ("DF", "MS", "GO", "MT")

# rep("blue", 4), -- Southesast ("MG", "RJ", "SP", "ES")

# rep("lightblue",3))-- South ("PR", "SC", "RS")

cor.radiation.1<- as.matrix(cor.radiation.1)

# to define the maximum cuting point (90%)

#boxplot(as.vector(cor.radiation.1))

#quantile(as.vector(cor.radiation.1), seq(0,1,0.10))

cor.radiation.1[cor.radiation.1<0.8880374] <- 0

network.radiation.1 <- graph_from_adjacency_matrix(cor.radiation.1, weighted=T, mode="upper", diag=F)

#plot(network, layout = layout.circle)

#par(mfrow=c(2,2), mar=c(1,1,1,1))

#plot(network.radiation.1, layout=layout.sphere, main="sphere")

#plot(network.radiation.1, layout=layout.circle, main="circle")

#plot(network.radiation.1, layout=layout.random, main="random")

#plot(network.radiation.1, layout=layout.fruchterman.reingold, main="fruchterman.reingold")

############

# BOX = 1261 

############

cor.radiation.2<- read.csv("C:\\\\Users\\\\toparo\\\\Desktop\\\\Florencio e Everaldo\\\\Networks\\\\radiation_1261.csv", sep=";", h=T)[,-1]

names(cor.radiation.2)<- c("SE", "PA", "MG", "RR", "DF", "MS", "MT", "PR", "SC", 

 "CE", "GO", "PB", "AP", "AL", "AM", "RN", "TO", "RS",

 "RO", "PE", "AC", "RJ", "BA", "MA", "SP", "PI", "ES")

cor.radiation.2<- as.matrix(cor.radiation.2)

cor.radiation.2[cor.radiation.2<0]<- 0

# to define the maximum cuting point (90%)

#boxplot(as.vector(cor.radiation.2))

#quantile(as.vector(cor.radiation.2), seq(0,1,0.10))

cor.radiation.2[cor.radiation.2< 0.58395693] <- 0

network.radiation.2 <- graph_from_adjacency_matrix(cor.radiation.2, weighted=T, mode="upper", diag=F)

#par(mfrow=c(2,2), mar=c(1,1,1,1))

#plot(network.radiation.2, layout=layout.sphere, main="sphere")

#plot(network.radiation.2, layout=layout.circle, main="circle")

#plot(network.radiation.2, layout=layout.random, main="random")

#plot(network.radiation.2, layout=layout.fruchterman.reingold, main="fruchterman.reingold")

############

# BOX = 3328 

############

cor.radiation.3<- read.csv("C:\\\\Users\\\\toparo\\\\Desktop\\\\Florencio e Everaldo\\\\Networks\\\\radiation_3328.csv", sep=";", h=T)[,-1]

names(cor.radiation.3)<- c("SE", "PA", "MG", "RR", "DF", "MS", "MT", "PR", "SC", 

 "CE", "GO", "PB", "AP", "AL", "AM", "RN", "TO", "RS",

 "RO", "PE", "AC", "RJ", "BA", "MA", "SP", "PI", "ES")

cor.radiation.3<- as.matrix(cor.radiation.3)

cor.radiation.3[cor.radiation.3<0]<- 0

# to define the maximum cuting point (90%)

#boxplot(as.vector(cor.radiation.3))

#quantile(as.vector(cor.radiation.3), seq(0,1,0.10))

cor.radiation.3[cor.radiation.3< 0.8373831] <- 0

network.radiation.3 <- graph_from_adjacency_matrix(cor.radiation.3, weighted=T, mode="upper", diag=F)

#par(mfrow=c(2,2), mar=c(1,1,1,1))

#plot(network.radiation.3, layout=layout.sphere, main="sphere")

#plot(network.radiation.3, layout=layout.circle, main="circle")

#plot(network.radiation.3, layout=layout.random, main="random")

#plot(network.radiation.3, layout=layout.fruchterman.reingold, main="fruchterman.reingold")

# =======================================

# https://briatte.github.io/ggnet/

# https://kateto.net/netscix2016.html

# install.packages("GGally")

#library(GGally)

#library(network)

#library(sna)

#library(ggplot2)

#########################################################

# to add collors accordingly to the Brazilian region

#col<- c(rep("green",9), rep("yellow", 7), rep("red", 4), rep("blue", 4), rep("lightblue",3))

col<- c("green", "yellow", "blue", "yellow", "red", "red", "red", "lightblue", "lightblue", 

 "green", "red", "green", "yellow", "green", "yellow", "green", "yellow", "lightblue",

 "yellow", "green", "yellow", "blue", "green", "green", "blue", "green", "blue")

p1<- ggnet2(network.radiation.1, label=T, color = col, fontface = "bold")

p2<- ggnet2(network.radiation.2, label=T, color = col, fontface = "bold")

p3<- ggnet2(network.radiation.3, label=T, color = col, fontface = "bold")

#library(gridExtra)

#library(grid)

#out<- grid.arrange(p1, p2, p3, nrow = 1, ncol=3)

#ggsave("Radiation.pdf", out)

#########################################################

b = theme(panel.background = element_rect(color = "grey50"))

gridExtra::grid.arrange(p1 + ggtitle("BOX = 13") + b,

 p2 + ggtitle("BOX = 1261") + b,

 p3 + ggtitle("BOX = 3328") + b,

 nrow = 1)

Note: For the DFA, DCCA and DCCA cross-correlation coefficient methods, I indicate the package in R language, available at:

https://cran.r-project.org/web/packages/DFA/index.html

The package with the methods was authored by the researchers: Florêncio Mendes Oliveira Filho and Paulo Canas Rodrigues. Patent: Computer Program. Registration number: BR512020002329-0, registration date: 10/26/2020, title: "DFA Package", Registration institution: INPI - National Institute of Industrial Property.

3. We note that the grant information provided in the 'Funding Information' and 'Financial Disclosure' sections do not match. When resubmitting, please ensure that you provide the correct scholarship numbers for the awards you received for your study in the 'Funding Information' section.

Answer: In case of acceptance of the manuscript by the journal, the institution of the first author (Centro Universitário Senai Cimatec, Engenharia da Computação, Salvador, Brazil) has undertaken to bear the costs of publication. Updates have been made to the acknowledgments section.

4. Thank you for stating the following in the acknowledgments section of your manuscript:

“Florêncio Mendes Oliveira Filho thanks National Council for Scientific and

Technological Development (CNPq 25/2021) ”PDJ scholarship'- Process: 150655/2022-3,

Brazilian agency. Paulo Canas Rodrigues thanks the financial support of

Scholarship from the National Council of Science and Technology (CNPq)

Productivity PQ-2”305852/2019-1”.

We have noticed that you have provided funding information that is not stated in your Funding Statement. However, funding information should not appear in the acknowledgments section or other areas of your manuscript. We will only post the funding information that appears in the Statement of Funding section of the online submission form.

Please remove any funding related text from the manuscript and let us know how you would like to update your Statement of Funding. Its Financing Statement currently reads as follows:

“The first author of the article is a professor at the Senai Cimatec University Center. In this sense, the Unitary Center will finance the publication costs. ”

Include your corrected statements in your cover letter; we will amend the online submission form on your behalf.

Answer: OK.

5. We noted in your submission details that a portion of your manuscript may have been presented or published elsewhere. [The research was submitted to Scientific Report (Peer Review Advisors) on 05 February 2022 UTC (Submission ID cd038b47-9e6e-42d6-9f25-5d2aac8cffd3). On 05/18/2022 we received the article for correction. On 05/11/2022 the article was rejected.

In the submission process to Scientific Report, the article adhered to the Research Square policy that was registered (DOI https://doi.org/10.21203/rs.3.rs-1330103/v1.).

After the rejection of "Scientific Reports – Nature" the article was submitted to Jornal do Clima and could not be continued (decision of the Editor-in-Chief Dr. Mingfang Ting) because our article is still being published in an environment linked to Scientific Reports, in this case the Research Square DOI.

In contact with Nusrat Tasneem (Editorial Support in Scientific Reports), the information given is that the DOI is no longer updated as soon as it is accepted by a journal. Please clarify whether publication was peer-reviewed and formally published. If this work was previously peer-reviewed and published, in the cover letter please provide the reason that this work does not constitute dual publication and should be included in the current manuscript.

Answer: The entire route of the manuscript submitted to the journals "Scientific Reports – Nature" and "Jornal do Clima" was detailed in the submission.

In the journal “Scientific Reports – Nature": the manuscript was peer-reviewed. With an indication to replace the methods with another completely out of context. It was not accepted by the authors.

As for “Jornal do Clima”: the manuscript did not advance because it adhered to Scientific Reports' Research Square policy.

6. We note that you have indicated that data from this study are available upon request. PLOS only allows data to be available upon request if there are legal or ethical restrictions on sharing data publicly. For more information on unacceptable data access restrictions, please see http://journals.plos.org/plosone/s/data-availability#loc-unacceptable-data-access-restrictions.

Answer: INPE data are public and available at: < https://www.gov.br/inpe/pt-br >.

Answer: A sample for verification is available at: < https://drive.google.com/file/d/1RAXMB2MTwFP3JEaXeDYEXVsyPEkM9vFm/view?usp=drive_link >.

7. We note that Figure 1S in your submission contain [map/satellite] images which may be copyrighted. All PLOS content is published under the Creative Commons Attribution License (CC BY 4.0), which means that the manuscript, images, and Supporting Information files will be freely available online, and any third party is permitted to access, download, copy, distribute, and use these materials in any way, even commercially, with proper attribution. For these reasons, we cannot publish previously copyrighted maps or satellite images created using proprietary data, such as Google software (Google Maps, Street View, and Earth). For more information, see our copyright guidelines: http://journals.plos.org/plosone/s/licenses-and-copyright.

a. You may seek permission from the original copyright holder of Figure 1S to publish the content specifically under the CC BY 4.0 license. 

Answer: We replace figure 1S as requested. The new figure, as well as the code, was created in R language. The code was made available in question 2 of this sequence. We now have a unique figure of his own authorship representing the regions of Brazil in the context of the manuscript.

Answer: OK

Additional Editor Comments:

In view of the comments from the reviewers, major revision of the manuscript would be required.

Reviewers' comments:

Reviewer's Responses to Questions

Comments to the Author

1. Is the manuscript technically sound, and do the data support the conclusions?

Reviewer #1: Yes

Reviewer #2: Partly

Answer #2: The entire manuscript is supported by a public database, available at < https://www.gov.br/inpe/pt-br > and its information can be used by any researcher. It is worth mentioning that INPE's public database contains information on environmental monitoring in Brazil, including satellite data and information on deforestation, fires, climate, among others. It is possible to access this database free of charge through the INPE website.

So that the evaluators can verify the veracity of the information, we will be making available a sample of the raw data. Available on the link, at: < https://drive.google.com/file/d/1RAXMB2MTwFP3JEaXeDYEXVsyPEkM9vFm/view?usp=drive_link >.

2. Has the statistical analysis been performed appropriately and rigorously?

Reviewer #1: Yes

Reviewer #2: Yes

Answer #1,#2: Thanks!

3. Have the authors made all data underlying the findings in their manuscript fully available?

Reviewer #1: Yes

Reviewer #2: No

Answer #2: We understand Plos and Reviewer #2's data policy. We emphasize that the paper uses public domain data, available at < https://www.gov.br/inpe/pt-br >. INPE's datasets contain information on environmental monitoring in Brazil, including satellite data and information on deforestation, fires, climate, among others.

4. Is the manuscript presented in an intelligible fashion and written in standard English?

Reviewer #1: Yes

Reviewer #2: No

Response #2: We agree and a new version of the text has been written. The changes are in red in the body of the manuscript. ('Revised manuscript with tracking changes')

5. Review Comments to the Author

Reviewer #1: Thank you for submitting your paper “Análise de redes de dados climáticos brasileiros com base no coeficiente de correlação cruzada DCCA” to the Journal of PLOS ONE.

In this paper, authors analyze four environmental and atmospheric variables, namely, wind speed, radiation,

temperature, and humidity, measured in 27 Weather Stations (the capital of each of the 26 Brazilian states plus the federal district).

The article is well written and well structured as a scientific text; the literature review is comprehensive and up to date. However, several issues need to be addressed properly before the paper is considered for publication.

Add the highlights

Add a table of nomenclature

Improve introduction and references

https://doi.org/10.1016/j.jobe.2021.103057

https://doi.org/10.1016/j.energy.2021.121641

https://doi.org/10.1016/j.enconman.2022.116554

https://doi.org/10.1016/j.jclepro.2023.137345

Figure 2S to 19S, Replace green vertical lines with black or different shades of gray for better readability.

In conclusion please explain how your work advances the present state of knowledge. Please, be more effective in conclusion paragraph. Critical discussion about the applicability and weak points of the proposed approach will make this manuscript much better.

Reviewer #2: relevant challenges that the world has to deal with.- what do you mean ‘relevant’ challenges ?

Line 73-74 deal with being the missing values imputed by the mean of that hour along

the days. –elaborate how it is done

However, [20] demonstrated that the DFA and DCCA methods are… make it more details are required 47 % missing means .. still DCCA works ? what exactly authors mean is not clear

Tabela 2.- Figura 1. The paper should be completely in English

Throughout the paper, the use of symbols should be checked For example kJ ; Line 223; Figure 2 etc in many parts

A proper connection between the three analysis performed is to be conveyed by a flowchart ;better clarity is required for line 130 onwards

IN Figures of DCCA, some of the lines are not visible ..proper color selection is required

The discussion section, especially last paragraph should be redrafted ..also what is the usefulness of the work/this graphical representations is not clear .. moreover the statements of climate change etc are vague

Answer #1:

Add highlights:

We understand that the highlight of this research is in several stages. We can quote:

(I) the treatment given to the data - a dense base that needed several adjustments.

(II) the interpretation of the Detrended fluctuation analysis - DFA method;

(III) and the mixed methodology of cross-correlation coefficient analysis – DCCA with Network analysis.

All these items were described in subsection for detailing and understanding.

Naming table:

In order to verify the Reviewer's suggestion, we scanned the manuscript for nomenclature. We verified that all abbreviations are mentioned in the body of the work (DFA, DCCA, INPE, ...) and that one more table would not improve its understanding or reading.

Improve the introduction and references:

About this item: The references were evaluated and added to the article. Regarding the introduction, improvements were made.

A description of the suggested articles:

1_[https://doi.org/10.1016/j.jobe.2021.103057] - This study provided a worldwide view of how comfort conditions inside a building can change according to external climatic conditions.

2_[https://doi.org/10.1016/j.energy.2021.121641] - This study concerns the dynamic thermal modeling over the years, up to 2080, of a multi-residential building located in Lecce, a city in the southeast of Italy, characterized by a warm Mediterranean summer climate.

3_[https://doi.org/10.1016/j.enconman.2022.116554] - This article investigates the effect of climate change on heat pumps, recognized worldwide as one of the most promising technologies for decarbonizing the construction sector.

4_[https://doi .org/10.1016/j.jclepro.2023.137345] – An important objective of this study is to determine whether

updates to building energy efficiency regulations from 2005 onwards, particularly for the envelope, will result in greater resilience of the envelope to climate change.

Figure 2S to 19S, Replace green vertical lines with black or different shades of gray for better readability.

All graphics in this manuscript were done manually using Origen 6.0 and Origen 2019b software. Before generating each graph, we tested several color possibilities on the vertical lines, as well as the background color. The dotted lines in blue refer to the scale relationship, when these are confronted with the black color the reference (scales) is not so aesthetically visible for reading. In grayscale, the graphic quality is lost, as well as the beauty of the graphics. We took the suggestion into account, but decided to keep the graphics as they are.

Below is an example with black vertical lines on the left and right in gray tone.

In conclusion please explain how your work advances the present state of knowledge. Please, be more effective in conclusion paragraph. Critical discussion about the applicability and weak points of the proposed approach will make this manuscript much better.

Here the reviewer cites two important points. First it refers to the conclusion and then the critical discussion and applicability.

About the conclusion:

The tool used to assess climate variables in this manuscript was the autocorrelation process with the DFA method and cross-correlation using the ρDCCA coefficient highlighted as a subsubsection. We cite two pioneering works in this type of analysis, the first was the article by Vassoler (Reference 44) and the article by Brito (Reference 44), both applied the coefficient to a small sample of cities and a smaller number of variables, not monitoring the entire the Brazilian territory as our manuscript is describing and also for a larger number of variables. And yet, with a hybrid methodology (Time Series + Networks). All this description is in the body of our manuscript and we can say yes, that this work has advanced a lot, since with these techniques, so far we do not have a study with this breadth in the literature here in Brazil.

About the critical discussion and applicability:

In the introduction already updated with new fonts suggested by Reviewer#1. The influence of climate variables points to interpretations such as comfort / discomfort, hot cities, thermal modeling, energy efficiency, climate change, ... in all of them a method was used to evaluate the studies. If verified in our discussion, you will see that we are showing contributions and possibilities of a different view of the tools and we affirm that the method / coefficient cited in our manuscript can be useful to understand and collaborate with various economic, agricultural, energy generation sector (hydraulic, wind and/or solar), sports and leisure activities and, above all, urban planning), among other possibilities.

We therefore understand that this suggestion is already included in the scope of the article.

Answer #2:

relevant challenges that the world has to deal with.- what do you mean 'relevant' challenges ?

We are referring to the practical consequences of climate change, still unknown by climatologists and meteorologists.

In Brazil;

Uncertainty: < 'Unknown territory': world is warmer and atypical El Niño has even more uncertain effects; understand | Environment | G1 (globo.com) >.

Consequences: < Climate change may cause loss of vegetation in 99% of the Caatinga by 2060, warns Unicamp study | Campinas and Region | G1 (globo.com) >.

In the world:

Environmental Forecast: < Monday, July 3, was Earth's hottest day on record, US agency says | Environment | G1 (globo.com) >.

We believe that these articles written in newspapers explain what we are talking about relevant challenges. The DFA method can be useful to study these trends.

the days. –elaborate how it is done

However, [20] demonstrated that the DFA and DCCA methods are… make it more details are required 47 % missing means .. still DCCA works ? what exactly authors mean is not clear.

About the missing values and 47% missing:

We cite the article (Reference 24), DCCA cross-correlation analysis in time-series with removed parts, to reference our treatment of the missing data.

This paper analyzes the effect of removing chunks (missing values) on long-range memory time series with the DFA method and the Untrendable Cross-Correlation Coefficient ρDCCA. From simulated series the results showed that up to 50% of the pieces removed, compared to the original time series, there is no change in the final results of the correlations. I understand that this article provides a good basis for our treatment of our series with missing data.

This information is in the body of the article in the section “Materials and methods”, subsection “Data collection and data organization”.

Table 2.- Figure 1. The paper should be completely in English

We checked! We saw no change in the language (English).

About KJ Unit:

This is the Global Radiation unit (KJ/M²) and is available at the base of the National Institute of Meteorology – INMET. Here is a Print of the base for verification. Picture below.

A proper connection between the three analysis performed is to be conveyed by a flowchart ;better clarity is required for line 130 onwards

One of the strategies adopted by the authors was to detail by subsection.

IN Figures of DCCA, some of the lines are not visible ..proper color selection is required

I commented in the previous item that due to the number of curves, the most appropriate color combination among the possibilities of the software we used to generate the figures, was the one presented in peper. Before answering Answer #2: I tested fundamental changes to the graphics, however we didn't notice any improvements. We remain with the original colors.

The discussion section, especially last paragraph should be redrafted ..also what is the usefulness of the work/this graphical representations is not clear .. moreover the statements of climate change etc are vague.

The discussion section, especially last paragraph should be redrafted

We do not understand the reason for the reformulation. Sorry, it wasn't clear.

also what is the usefulness of the work/this graphical representations is not clear .. moreover the statements of climate change etc are vague.

As described in the body of the article, our research aims to evaluate 4 environmental and atmospheric variables (global radiation, temperature, humidity and wind speed) for 27 meteorological stations in Brazil. For that, we used the DFA method and the ρDCCA cross-correlation coefficient.

With a wealth of graphs and tables we show the influence of scale for various regions and how these environmental variables correlate for various time scales.

Technically, in graphics, it refers to the influence of methods.

6. PLOS authors have the option to publish the peer review history of their article (what does this mean?). If published, this will include your full peer review and any attached files.

Do you want your identity to be public for this peer review? For information about this choice, including consent withdrawal, please see our Privacy Policy.

Reviewer #1: No

Reviewer #2: No

---

## [Editor Report · Decision Letter 1]

16 Aug 2023

Networks Analysis of Brazilian Climate Data based on the

DCCA cross-correlation coefficient

PONE-D-23-12144R1

Dear Dr. Oliveira Filho,

We’re pleased to inform you that your manuscript has been judged scientifically suitable for publication and will be formally accepted for publication once it meets all outstanding technical requirements.

Kind regards,

Pak Wai Chan

Academic Editor

PLOS ONE
---

## [Editor Report · Acceptance letter]

7 Sep 2023

PONE-D-23-12144R1 

Networks Analysis of Brazilian Climate Data based on the DCCA cross-correlation coefficient 

Dear Dr. Oliveira Filho:

I'm pleased to inform you that your manuscript has been deemed suitable for publication in PLOS ONE. Congratulations! Your manuscript is now with our production department. 

Kind regards, 

on behalf of

Dr. Pak Wai Chan 

Academic Editor

PLOS ONE